# The Effect of Varying Interval Definitions on the Prevalence of SARS-CoV-2 Reinfections: A Retrospective Cross-Sectional Cohort Study

**DOI:** 10.3390/diagnostics12030719

**Published:** 2022-03-15

**Authors:** Sjoerd M. Euser, Tieme Weenink, Jan M. Prins, Milly Haverkort, Irene Manders, Steven van Lelyveld, Bjorn L. Herpers, Jan Sinnige, Jayant Kalpoe, Dominic Snijders, James Cohen Stuart, Fred Slijkerman Megelink, Erik Kapteijns, Jeroen den Boer, Alex Wagemakers, Dennis Souverein

**Affiliations:** 1Regional Public Health Laboratory Kennemerland, 2035 RC Haarlem, The Netherlands; s.euser@streeklabhaarlem.nl (S.M.E.); p.t.weenink@gmail.com (T.W.); i.manders@streeklabhaarlem.nl (I.M.); b.herpers@streeklabhaarlem.nl (B.L.H.); j.sinnige@streeklabhaarlem.nl (J.S.); j.kalpoe@streeklabhaarlem.nl (J.K.); j.denboer@streeklabhaarlem.nl (J.d.B.); a.wagemakers@streeklabhaarlem.nl (A.W.); 2Department of Internal Medicine, Division of Infectious Diseases, Amsterdam UMC, University of Amsterdam, Meibergdreef 9, 1105 AZ Amsterdam, The Netherlands; j.m.prins@amsterdamumc.nl; 3Public Health Service Kennemerland, 2015 CK Haarlem, The Netherlands; mhaverkort@vrk.nl; 4Spaarne Gasthuis, Hoofddorp, 2134 TM Haarlem, The Netherlands; s.van.lelyveld@spaarnegasthuis.nl (S.v.L.); dsnijders@spaarnegasthuis.nl (D.S.); 5Noordwest Ziekenhuisgroep, 1815 JD Alkmaar, The Netherlands; j.w.t.cohenstuart@nwz.nl; 6Public Health Service Hollands Noorden, 1823 DL Alkmaar, The Netherlands; fslijkerman@ggdhn.nl; 7Rode Kruis Ziekenhuis, 1942 LE Beverwijk, The Netherlands; ekapteijns@rkz.nl

**Keywords:** SARS-CoV-2, re-infection, COVID-19, Cp-value

## Abstract

Background: We assessed the SARS-CoV-2 reinfection rate in a large patient cohort, and evaluated the effect of varying time intervals between two positive tests on assumed reinfection rates using viral load data. Methods: All positive SARS-CoV-2 samples collected between 1 March 2020 and 1 August 2021 from a laboratory in the region Kennemerland, the Netherlands, were included. The reinfection rate was analyzed using different time intervals between two positive tests varying between 2 and 16 weeks. SARS-CoV-2 PCR crossing point (Cp) values were used to estimate viral loads. Results: In total, 679,513 samples were analyzed, of which 53,366 tests (7.9%) were SARS-CoV-2 positive. The number of reinfections varied between 260 (0.52%) for an interval of 2 weeks, 89 (0.19%) for 4 weeks, 52 (0.11%) for 8 weeks, and 37 (0.09%) for a minimum interval of 16 weeks between positive tests. The median Cp-value (IQR) in the second positive samples decreased when a longer interval was chosen, but stabilized from week 8 onwards. Conclusions: Although the calculated reinfection prevalence was relatively low (0.11% for the 8-week time interval), choosing a different minimum interval between two positive tests resulted in major differences in reinfection rates. As reinfection Cp-values stabilized after 8 weeks, we hypothesize this interval to best reflect novel infection rather than persistent shedding.

## 1. Introduction

Since the start of the COVID-19 pandemic in early 2020, SARS-CoV-2 was first expected to induce a monophasic disease with at least transient immunity [1]. However, several reports of SARS-CoV-2 reinfections have been reported since [2], which raised critical questions about how well a first infection protects against reinfection [3,4,5,6]. In addition, the occurrence of SARS-CoV-2 reinfection has implications for epidemiological modelling and public health policies with respect to the distribution of (booster) vaccines, and the influence of circulating new genetic variants of SARS-CoV-2 [7,8,9].

Studies conducted to address the incidence of SARS-CoV-2 reinfections are scarce, with most studies reporting single cases or small case series [6,7,10]. The few larger cohort studies that assessed the incidence of reinfection with SARS-CoV-2 estimated a reinfection rate ranging from 0.10% to 0.26% during varying follow-up periods of 2–9 months [11,12,13]. The interpretation of these reports of SARS-CoV-2 reinfections is hampered by the methodological inconsistencies between studies. For instance, to distinct true reinfections from prolonged viral shedding, viral loads of the first and second positive test may be helpful. However, SARS-CoV-2 viral loads in respiratory samples lack comparability of Ct- or Cp-values derived from different laboratories, as these are assay- and method-specific [14]. This issue complicates the evaluation of SARS-CoV-2 viral loads in respiratory samples derived from potential reinfection patients when multiple laboratories are involved in analyzing these samples, which is often the case [13]. It should also be noted that most studies use different definitions of reinfection: though some choose a stringent definition including the requirement of a negative SARS-CoV-2 PCR result between two positive tests, reporting of COVID-19-related symptoms during both episodes, a minimal viral load, and evidence of genotypic variance between the two viral strains [7], others use only a selection of these [15]. The major difference between studies, though, seems to be the chosen minimum time interval between two positive tests, which ranges from 8 to 82 days between studies [2,7], and which is likely to have a large influence on the reported prevalence of reinfection. One of the reasons for including a time interval criterion is to exclude patients who show prolonged viral shedding, which is thought to disappear after 28 days in most COVID-19 cases [16], but has occasionally been demonstrated up to 63 days after symptom onset [17].

The aim of this study was to assess the incidence of reinfection with SARS-CoV-2, to evaluate the effect of varying time intervals between two positive tests on assumed reinfection rates, and to incorporate viral load data of both the first and second positive test in these analyses in a large cohort of SARS-CoV-2-positive patients tested in a single large regional laboratory in the Netherlands. Thus, we set out to provide an evidence-based time interval to be incorporated in the definition of SARS-CoV-2 reinfection.

## 2. Methods

### 2.1. Setting, Study Design and Participants

The Regional Public Health Laboratory Kennemerland, Haarlem, the Netherlands, performs SARS-CoV-2 RT-PCR testing for over 800,000 inhabitants, including health care workers (HCW), patients of four large teaching hospitals, patients of more than 600 GPs, 90 nursing home organizations, and those who are tested because of the presence of symptoms in public health testing facilities set up by the Public Health Services Kennemerland (PHS Kennemerland) and Hollands Noorden (PHS Hollands Noorden). Data from all SARS-CoV-2 RT-PCR results from nasopharyngeal (NP), oropharyngeal (OP), and combined swabs collected between 1 March 2020 and 1 August 2021 were analyzed in the present study. For the patients with more than one positive test result obtained at least one week apart, we retrieved the age, the dates of both positive tests, and the Cp-value of both positive tests from the electronic patient records of the Public Health Laboratory Kennemerland.

### 2.2. Assays

The SARS-CoV-2 RT-PCR used, which is based on the presence of the E-gene [18], was carried out after a lysis step of the samples. For all swabs, crossing-point (Cp)-values were calculated on Lightcycler 480 1.5.1 software (Roche diagnostics, Basel, Switzerland). We used sequencing data of the Dutch National Institute for Public Health and the Environment to estimate the circulation of specific viral variants in the population during our study period [19].

### 2.3. Statistical Analysis

Prevalence of reinfection was calculated using four different time intervals between two positive SARS-CoV-2 tests: 2 weeks; 4 weeks; 8 weeks; and 16 weeks. Continuous variables were presented as median with interquartile ranges (IQR), and categorical variables as numbers with percentages. Statistical analyses were performed with R and RStudio (R v 4.0.3), with packages tidyverse and tidymodels.

## 3. Results

In total, 679,513 samples were collected from 473,411 unique patients, of which 53,366 (7.9%) tests derived from 51,484 unique patients were positive for SARS-CoV-2. In Figure 1, the weekly number of positive tests is presented, as well the cumulative number of reinfections for each chosen time interval. In addition, the proportion of circulating viral variants in the Netherlands during the study period is shown (in background colors). These data show a gradual increase in the number of reinfections over time, with an acceleration in the last weeks when the Delta variant became dominant.

Each colored line corresponds to one specific minimum time interval between two positive tests (2–16 weeks). Bars represent the number of reported SARS-CoV-2 positive tests per week. Background colors show the proportion of the circulating viral variant during the study period.

Table 1 shows, per time interval chosen, the number of reinfections, median age of patients, and Cp-values (indicating viral load) of the first and second positive tests. The number of reinfections decreased from 260 (0.52%) reinfections for a minimum interval of 2 weeks, to 37 (0.09%) reinfections for an interval of 16 weeks. The median age (IQR) of the patients decreased considerably when a longer time interval was chosen: 69.0 (33.5) years for the 2-week interval vs. 24.0 (30.0) years for the 16-week interval. The median Cp-value (IQR) in the second positive samples decreased (reflecting higher viral loads) when a longer interval was chosen: Cp-value 34.1 (6.1) for the 2-week interval vs. 28.7 (6.5) for the 16-week interval.

In Figure 2, the median Cp-values are presented for the first and second positive RT-PCR tests according to reinfection definitions (ranging from 1 to 20 weeks). Whereas 684 “reinfection” cases were included using a minimum of 1 week between positive tests, this number decreased when using stricter definitions (Appendix A). The first positive PCR had a similar median Cp-value in all case definitions. However, the second positive PCR showed a high median Cp-value when a short interval between positive PCRs was used to define reinfection, suggesting a significant contribution from the detection of persistent shedding (with low viral loads). The Cp-value of the second PCR stabilized when at least 8 weeks between positives was used to define cases, suggesting persistent shedding no longer explained the second positive PCR. Reinfections defined by at least 8 weeks between positive PCRs consistently had higher median Cp-values at the time of reinfection compared to the first episode, indicating lower viral loads than during the first infection.

The solid line corresponds to the median Cp-value of the first positive test of reinfection patients for increasing minimum time intervals between two positive tests (ranging from 1 to 20 weeks). The dotted line corresponds to the median Cp-value of the second positive test of these reinfection patients.

## 4. Discussion

Our study shows that it is necessary to establish a consensus on the definition of SARS-CoV-2 reinfection, as this has implications for epidemiological modelling and public health policies [13]. Depending on the time interval chosen, the percentage of reinfection varied between 0.09% and 0.52%. The calculated reinfection prevalence for the 8-week time interval (0.11%) is in line with previous studies with comparable follow-up [11,12,13].

In previous studies, a standardized time interval between two positive tests to define reinfections was lacking [7,11,12,13,15], and a wide range of reinfection rates were reported [11,12,13]. This could possibly (at least partly) be explained by a phenomenon called “prolonged shedding”. In some patients, traces of the viral RNA are detectable for a relatively long period after onset of symptoms, sometimes even when the symptoms have disappeared [17]. Our data showed increasing viral loads in the second test sample when the time interval to define a reinfection was extended, with a plateau in median viral load of an approximate Cp-value of 28.7 reached at 8 weeks since the first positive test. This implicates that the preferred time interval between two positive tests should be at least 8 weeks, in line with previous observations [13], and as suggested by international guidelines [20,21,22].

One of the limitations of our study concerns the possibility of underreporting of reinfections in our population. Large scale public health test facilities became available in the Netherlands in May 2020. During the first COVID-19 wave in the Netherlands, testing was mainly available for health care workers [23]. Therefore, the majority of the SARS-CoV-2 infections that occurred between the start of the pandemic in February 2020 and June 2020 were not registered, and reinfections were not recognized if patients were tested positive in the following months. Another potential reason for underreporting is the fact that secondary infections might be less often reported. Patients who recover from a primary SARS-CoV-2 infection may be less likely to have themselves tested when they develop respiratory symptoms again, assuming that they are immune for SARS-CoV-2. Finally, it should be noted that not all of the collected samples in the regions Kennemerland and Hollands Noorden were analyzed at the Regional Public Health Laboratory Kennemerland and registered in our database. This could have led to an underestimation of the reinfection prevalence in our population. Also, patients might have tested positive outside our region. Although this may have affected the reinfection rates in our study (and comparability with other studies), it does not affect our finding that the preferred time interval between two positive tests should be at least 8 weeks.

Unfortunately, no clinical details were available for the cases, including data on immunocompromising underlying conditions of participants or their vaccination status. This limited the possibility to assess the influence of these factors on the occurrence of re-infection in individual patients, and to accurately distinguish between prolonged shedding and relapse of re-infection cases. In addition, no sequencing results were available for individual samples, making it difficult to establish full certainty of reinfection. However, as (re-)infections can be asymptomatic, clinical details alone do not provide sufficient evidence for reinfection. Sequencing can be performed in individual cases to ascertain that a reinfection is caused by a different variant. This was, however, not feasible on a large scale in our public health setting. However, to the best of our knowledge, we provide the largest dataset on reinfections and Cp-values (or viral loads) performed with a single PCR-technique, enabling comparison between samples. Also, we aimed to establish a simple time-interval-based definition that can be used in all labs that perform SARS-CoV-2 PCR testing, regardless of the availability of rapid sequencing or clinical information.

In conclusion, it is of major importance to establish a consensus on the definition of SARS-CoV-2 reinfections, as the influence of waning immunity, and the circulating and spreading of new SARS-CoV-2 variants on reinfection rates can only be valued when a uniform definition is used between studies. Based on our data, we believe that an 8-week period between two positive PCRs can be used as a definition for reinfection cases in case sequencing, and conclusive clinical information is lacking, thus providing an epidemiological tool for future studies.

## Figures and Tables

**Figure 1 diagnostics-12-00719-f001:**
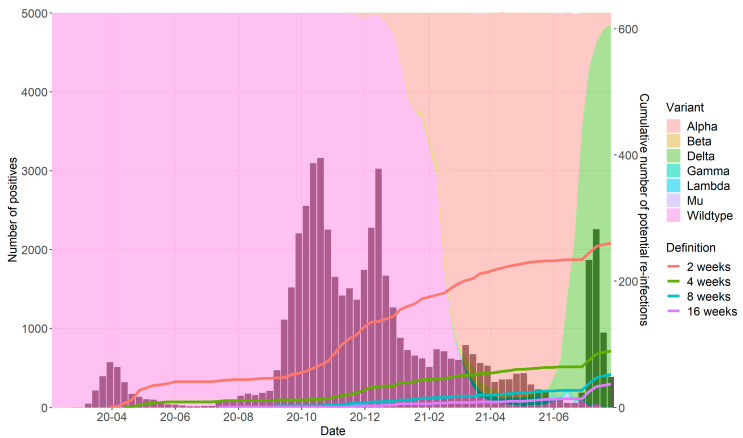
Cumulative number of reinfections according to different time intervals between two positive tests.

**Figure 2 diagnostics-12-00719-f002:**
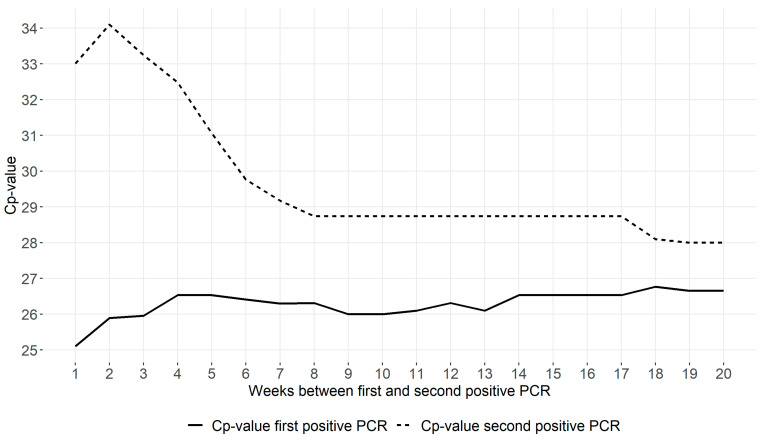
Median Cp-values of first and second positive RT-PCR tests of reinfection cases according to different time intervals between two positive tests.

**Table 1 diagnostics-12-00719-t001:** Number of reinfections according to different definitions.

Definition	Number of Reinfections (%)	Median Age [IQR] (Years)	Median Cp-Value [IQR] First Positive Test	Median Cp-Value [IQR] Second Positive Test
2 weeks	260 (0.52%)	69.0 [33.5]	25.9 [7.5]	34.1 [6.1]
4 weeks	89 (0.19%)	53.0 [50.0]	26.5 [8.5]	32.5 [8.1]
8 weeks	52 (0.11%)	30.0 [34.0]	26.3 [8.8]	28.7 [8.1]
16 weeks	37 (0.09%)	24.0 [30.0]	26.5 [8.8]	28.7 [6.5]

Data are presented as number (%) or median [IQR]. IQR = interquartile range.

## Data Availability

Data are available from the authors upon reasonable request.

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
