# Peer review of "The Effect of Varying Interval Definitions on the Prevalence of SARS-CoV-2 Reinfections: A Retrospective Cross-Sectional Cohort Study"

_diagnostics, 2022, doi:10.3390/diagnostics12030719_

Round 1

Reviewer 1 Report

Dear Editors,

Thank you for the opportunity to review the manuscript titled “ The effect of varying interval definitions on the prevalence of SARS-CoV-2 reinfections: a retrospective cross-sectional cohort study” by Sjoerd M. Euser et al. The manuscript is well written and the topic is interesting. The authors aim to provide an evidence-based time interval to be incorporated in the definition of SARS-CoV-2 reinfection and in my opinion they partly succeed, and the findings could be helpfull in assisting future research design.

Here are some specific comments.

The study period needs to be precisely defined in the abstract. The study periods mentioned in the abstract and methods sections differ. Explanations  or corrections are needed.

Immunocompromised patients may have prolonged shedding periods. Are there any available data about immunocompromised patients in this cohort? If not, this serious limitation should be included in the limitations section, together with the lack of clinical and genetic sequencing data, that makes the accurate classification of cases as prolonged shedding, relapse, of re-infection impossible in the present study. 

Author Response

Dear Editors,

Thank you for the opportunity to review the manuscript titled “ The effect of varying interval definitions on the prevalence of SARS-CoV-2 reinfections: a retrospective cross-sectional cohort study” by Sjoerd M. Euser et al. The manuscript is well written and the topic is interesting. The authors aim to provide an evidence-based time interval to be incorporated in the definition of SARS-CoV-2 reinfection and in my opinion they partly succeed, and the findings could be helpfull in assisting future research design.

Here are some specific comments.

The study period needs to be precisely defined in the abstract. The study periods mentioned in the abstract and methods sections differ. Explanations  or corrections are needed.

We fully agree with the reviewer and have revised the manuscript accordingly:

“Methods: All positive SARS-CoV-2 samples collected between March 1, 2020 and August 1, 2021 from a laboratory in the region Kennemerland, the Netherlands, were included.”

Immunocompromised patients may have prolonged shedding periods. Are there any available data about immunocompromised patients in this cohort? If not, this serious limitation should be included in the limitations section, together with the lack of clinical and genetic sequencing data, that makes the accurate classification of cases as prolonged shedding, relapse, of re-infection impossible in the present study. 

We agree with the reviewer and have now included a more extensive limitation section in the Discussion:

“Unfortunately, no clinical details were available for the cases, including data on immunocompromising underlying conditions of participants or their vaccination status. This limited the possibility to assess the influence of these factors on the occurrence of re-infection in individual patients and to accurately distinguish between prolonged shedding, relapse, of re-infection cases. In addition, no sequencing results were available for individual samples, making it difficult to establish full certainty of reinfection.”

Reviewer 2 Report

In this perspective, Euser et al discuss the effect of varying time intervals on the prevalence of SARS-CoV-2 reinfections. They calculated the prevalence of reinfection using RT-PCR using four different time intervals between two positive SARS-CoV-2 tests: 2 weeks; 4 weeks; 8 weeks; and 16 weeks. They conclude the calculated reinfection prevalence for the 8 week time interval (0.11%) is in line with previous studies, where viral shedding from previous infection is almost absent.

The study could be considered for publication after few points have been addressed.

  1. Did vaccination status play a role in the reinfection prevalence in this study? Does vaccination status affect the time period required for viral shedding?
  2. What are better ways to detect reinfections that happen during the period of viral shedding from the first infection?

Author Response

In this perspective, Euser et al discuss the effect of varying time intervals on the prevalence of SARS-CoV-2 reinfections. They calculated the prevalence of reinfection using RT-PCR using four different time intervals between two positive SARS-CoV-2 tests: 2 weeks; 4 weeks; 8 weeks; and 16 weeks. They conclude the calculated reinfection prevalence for the 8 week time interval (0.11%) is in line with previous studies, where viral shedding from previous infection is almost absent.

The study could be considered for publication after few points have been addressed.

  1. Did vaccination status play a role in the reinfection prevalence in this study? Does vaccination status affect the time period required for viral shedding?

We do not have the availability over clinical details for the cases described in our manuscript, and have now added an additional comment on this limitation in the revised manuscript:

“Unfortunately, no clinical details were available for the cases, including data on immunocompromising underlying conditions of participants or their vaccination status. This limited the possibility to assess the influence of these factors on the occurrence of re-infection in individual patients and to accurately distinguish between prolonged shedding, relapse, of re-infection cases.

  1. What are better ways to detect reinfections that happen during the period of viral shedding from the first infection?

Thank you for this interesting question. If one would not include the requirement of a negative SARS-CoV-2 PCR result between two positive tests as part of the definition of re-infection (which is unlikely to occur during the period of viral shedding from the first infection), we suppose that genetic analyses using sequencing data of the consecutive positive SARS-CoV-2 samples is essential to detect reinfections and distinguish between positive samples as a consequence of prolonged shedding from true reinfections (where different sequence results are found).
